# Drought Offsets the Potential Effects of Nitrogen Addition on Soil Respiration and Organic Carbon in Model Subtropical Forests

Yu-lin Zhu [1], Xue-ping Lin [1], Yun-peng Huang [2], Xing-hao Tang [2], Xiong Fang [3],* and Zhi-gang Yi [1]

[1]  College of Resources and Environment, Fujian Agriculture and Forestry University, Fuzhou 350002, China
[2]  Fujian Academy of Forestry Science, Fuzhou 350012, China
[3]  College of Land Resources and Environment, Jiangxi Agricultural University, Nanchang 330045, China
*   Correspondence: fangxiong_1103@126.com

**Abstract:** Nitrogen (N) deposition is increasingly aggravating and has significant impact on the processes of forest soil carbon (C) cycling. However, how N deposition affects forest soil C cycling processes in the scenario of future drought-frequent climate is still unclear. Therefore, we conducted a 2.5-year experiment at two levels of N addition treatments (control and N addition) and three levels of moisture (well-watered: ca. 80% of field capacity, moderate drought: ca. 60% of field capacity, severe drought: ca. 40% of field capacity) to investigate the impact of N addition, drought, and their interaction on soil respiration (Rs) and soil organic carbon (SOC) content. The results showed that N addition significantly increased Rs and SOC content, and severe drought decreased Rs and SOC content. In a well-watered condition, N addition significantly increased annual mean Rs, but in moderate drought and severe drought condition, N addition did not obviously affect Rs. In the control group, severe drought significantly decreased annual mean Rs by 61.5%, and decreased SOC content in 0–10 cm and in 10–20 cm by 3.0% and 1.6%, respectively. However, in the N addition group, moderate drought and severe drought significantly decreased annual mean Rs by 27.6% and 70.5%, respectively. Meanwhile, compared to the well-watered condition, severe drought significantly decreased SOC content in 0–10 cm and in 10–20 cm by 12.4% and 11.9% in the N addition group, respectively. Severe drought also decreased aboveground and belowground biomass, fine root biomass, MBC, and specific respiration in N addition group. The Rs and SOC content were positively correlated with aboveground biomass, belowground biomass, and fine root biomass. These results suggest that under future global change scenarios, severe drought might offset the promotive effects of N deposition on soil respiration and C sequestration in the young subtropical forest. Moreover, the N deposition may enhance the suppressive effect of drought on soil respiration and C sequestration in the future.

**Keywords:** nitrogen deposition; drought; soil respiration; soil carbon; subtropical forest





## 1. Introduction

Forest soil is the largest carbon (C) pool in terrestrial ecosystems [1,2]. As one of the main pathways for soil and atmospheric C flow, soil respiration (Rs) plays a crucial role in regulating soil C pools and atmospheric $CO_2$ concentrations [3]. Recently, the excessive use of fossil fuels and agricultural fertilizers has resulted in a massive influx of active nitrogen (N) into the atmosphere [4,5], making N deposition a major cause of global change. Thus, whether and how N deposition changes the process of carbon cycling and sequestration in forest soil has garnered research interest, but there is still much uncertainty. This is partly due to the influence of climatic regions, vegetation types, or other global change factors (e.g., drought, etc.) [6,7]. For example, due to global warming and atmospheric circulation anomaly, the severe drought frequently occurs on a global scale [8], which also has significant influence on the soil carbon cycling [9,10]. However, there is no clear

evidence how forest Rs is affected by combined N deposition with drought, and further influencing forest soil carbon sequestration. Therefore, understanding the comprehensive effects of N deposition and drought on soil respiration and the potential implications on soil C sequestration in forests is imperative for predicting soil C sink potential under future climate scenarios.

Soil nitrogen and soil moisture availability are important factors affecting soil respiration and C sequestration [11,12]. On the one hand, they can affect plant-derived C input or roots respiration by affecting plant growth [13,14]. On the other hand, they can also influence soil respiration by affecting microbial decomposition processes of soil C [15,16]. Numerous experimental studies have been conducted in order to investigate the effect of N deposition on Rs and soil carbon [17,18]. Additionally, most of these pieces of research reported that N deposition may decrease soil respiration [19,20], further increasing the accumulation of soil organic carbon [21]. However, there are also controversies about the effect of N deposition on Rs and C capture in different climate zones and ecosystems. For example, Bowden et al. [15] found that N addition may decrease Rs in temperate forest, but Xu and Wan [22] found N addition had positive effect on Rs in a semiarid grassland. Meanwhile, Mack et al. [23] reported that long-term nutrient fertilization reduced carbon storage in arctic tundra, but Pregitzer et al. [21] showed that N addition may increase soil C sequestration in northern temperate forest. These differences may be related to soil moisture changes. In general, reduced soil moisture limits soil N availability and mobility, which, in turn, would affect soil Rs and C sequestration under N addition conditions [24]. However, there were few consensuses on how N deposition, drought, and their interaction affect soil respiration now. Thus, it is necessary to study the impacts of N deposition on Rs and soil C under drought conditions in the forest.

Over the past decades, several N addition and drought interaction experiments have investigated the comprehensive impact of N and moisture on Rs and C sequestration [25,26]. They demonstrated that the impact of N on Rs might be regulated by soil moisture [27]. First, suitable soil moisture is in favor of the mobility of exogenous N, thus stimulating plant root growth [28] and further promoting root respiration. In addition, both N addition and suitable soil moisture can increase litter and root-derived C input [29]. Thus, soil heterotrophic respiration could be regulated by changes in belowground C supply [25,30]. This may be due to altered soil microbial biomass, the composition of microbial community, or microbial metabolic activity [31]. Meanwhile, soil moisture is an important factor in maintaining microbial activity and metabolic processes [11,16]. Therefore, the effect of N on microbial decomposition will also be influenced by soil moisture. Currently, there have been some studies on the effects of N addition and drought interactions on soil respiration, most of these studies have been focused on grassland [29,32]. However, due to technical or cost factor, there is lack of studies to investigate the effect of N and drought interaction on forest soil respiration and soil C in the tropical and subtropical forests.

Tropical and subtropical forests contain approximately a quarter of carbon of the terrestrial biosphere and have important influences on C cycling and biodiversity on the global scale [33]. Recently, the subtropical forests are constantly accumulating C in large enough quantities to influence global C budget and C balance [34]. Meanwhile, subtropical area in China is the third largest nitrogen deposition area in the world [35,36], and drought has become more severe and frequent in recent years [8]. However, there is limited understanding about the potential effects of N and drought interaction on Rs and C sequestration in subtropical forests. Therefore, we conducted a 2.5-year manipulation experiment, using two N addition rates and three watering levels in order to examine the effects of N deposition, drought, and their interaction on Rs, SOC content, above and belowground biomass, soil microbial carbon, and other relative parameters in model subtropical forests in southern China. The main hypotheses are: (1) N addition would inhibit soil respiration, but drought may magnify this inhibitory effect in the young model forests; (2) N addition may stimulate SOC accumulation by reducing the process of soil

C decomposition, while drought may inhibit this promotion by reducing plant-derived C input.

## 2. Material and Methods

### 2.1. Experimental Sign

This experiment was conducted at Fujian Agriculture and Forestry University, located in Fujian province in south-eastern China (26°5′9″ N, 119°14′19″ E). The climate is subtropical monsoon, with mean rainfall and temperature at 1700 mm and 21.5 °C, respectively. The growing season is from April to October, and non-growing season is from November to March.

In October 2017, we built 18 square plots (length × width × height; 1 × 1 × 0.6 m) in an open-air greenhouse. One drainage port (2 cm) at the bottom of each square pond was connected to a polyvinyl chloride tube to collect the soil leachate. Two different soil layers (0–25, 25–50) were collected in a nearby secondary natural forest in November 2017, mixed separately after picking out coarse roots and stones, and then filled into the corresponding plot (0.5 m depth). The soil pH, $NH^+_4$-N, and $NO^-_3$-N were 6.09 ± 0.08, 5.62 ± 0.77, 2.20 ± 0.51 mg kg$^{-1}$, respectively. After approximately 2 months of soil natural sinking, four local typical plants species were transplanted into each plot randomly [37]. Four subtropical tree species, *Pinus massoniana* Lamb., *Cunninghamia lanceolata* (Lamb) Hook., *Schima superba* Gardn. Et Champ. and *Ormosia pinnata* (Lour.) Merr., were selected for the present study.

In April 2018, the experiment was initiated with two levels of N addition treatments, which was assigned to control (0 kg N ha$^{-1}$ y$^{-1}$) and N addition (80 kg N ha$^{-1}$ y$^{-1}$). Within each N addition treatment, the plots were divided into three moisture gradients, well-watered treatment (ca. 80% of field capacity), moderate drought (ca. 60% of field capacity), and severe drought (ca. 40% of field capacity). The designed soil moisture was calculated from soil bulk density and the volume of the square plot. Soil temperature and moisture were measured using a soil temperature and moisture meter (TR-6, Shunkeda, Hong Kong, China). From April 2018 to August 2020, soil temperature was measured and soil moisture was detected once in 3–5 days. Ammonium nitrate ($NH_4NO_3$) was used as the nitrogen source, 1.905 g of ammonium nitrate and 5 L of water were mixed and the solution was sprayed once a month. The control group received equal amounts of deionized water.

### 2.2. Soil Respiration Measurements

One PVC soil collar (11 cm in diameter, 8 cm deep) was buried 5 cm into the soil in each plot in August 2018. Gas samples were collected using the static chamber method, and the soil collar was sealed with a PVC jar each time when the gas sample was collected. Before collecting the gas, the headspace is flushed with the free air for 60 s to ensure initial $CO_2$ standardization [38]. An amount of 30 mL of air was drawn from headspace into aluminum foil bag to measure the initial $CO_2$ concentration. The soil collars were sealed for 30 min, after which $CO_2$ measurements were repeated. Gas samples were analyzed by chromatography (7890B, Agilent, Santa Clara, CA, USA). According to Xing et al. [39], the soil respiration rate (*Rs*) was calculated by the Equation (1):

$$Rs = \rho \times V/A \times \Delta c / \Delta t \times 273/(273 + T)/M \qquad (1)$$

where $\rho$ is the density of $CO_2$ under standardized state; *V* is the volume of seal PVC jars; *A* represents the area from which $CO_2$ emitted in to the seal PVC jars; $\Delta c / \Delta t$ is the average $CO_2$ concentration difference per second; *T* represent environmental temperature in the open-air greenhouse; and *M* represent the molar mass of $CO_2$. According to Fang et al. [40], the cumulative amount of soil respiration rate ($C_m$) was calculated by following equation:

$$C_m = C_{m-1} + (Rs_p + Rs_{p-1})/2 \times (D\text{-}D_{-1}) \qquad (2)$$

where *Rs*, p, and D are soil respiration rate per month, experimental period, and time (day), respectively. Annual mean soil respiration rates were calculated from January to December 2019.

### 2.3. Aboveground Biomass and Belowground Biomass

To measure biomass, trees in different treatments were harvested in August 2020. Those trees were divided into roots, stems, and leaves. Samples were oven-dried at 65 °C before being weighed. The biomass of each plot was calculated with four trees in each plot. The aboveground biomass was the sum of stem and leaf biomass, and the belowground biomass was the roots biomass. Fine roots collected during sieving was used for fine root biomass.

### 2.4. Soil Organic Carbon, Microbial Carbon, and Dissolved Organic Carbon

Soil samples were collected in August 2020. Five soil cores (4.5 cm in diameter) within each plot were randomly collected at 0–10 cm and 10–20 cm depth. According to the different depths of the collected soil, soil samples were mixed separately. All samples were passed through a 2 mm sieve to pick out fine roots, litter, and plants residues. Then, each soil sample was divided into two sub-samples: the first one was air-dried for the analysis of basic soil properties, and the second one was stored at 4 °C for later soil microbial biomass determination.

Soil pH was measured using a 1:2.5 ratio of soil to deionized water volume. SOC content and total N were measured by using Elemental Analyzer (EA3000, Euro Vector, Pavia, Italy). Soil ammonium nitrogen ($NH_4^+$-N) and nitrate nitrogen ($NO_3^-$-N) were determined by potassium chloride leaching-UV spectrophotometric colorimetric method [41]. Soil available phosphorus (AP) was determined by 0.5 mol $L^{-1}$ sodium bicarbonate leachingmolybdenum antimony anti-colorimetric method.

Chloroform fumigation-exaction method was used to measure soil microbial biomass C (MBC). The MBC was extracted with potassium sulfate on both fumigated and unfumigated soil; conversion coefficient was 0.45 for MBC [42]. The C content of the extract was determined using a TOC analyzer (Shimadzu Corp., Kyoto, Japan). MBC content was calculated by the difference between the C content of fumigated and the unfumigated soil. Specific respiration was calculated as dividing the cumulative amount of soil respiration by MBC content. The extractable C from the unfumigated samples were used to calculate dissolved organic carbon (DOC) content.

### 2.5. Data Analysis

Before data analysis, logarithmic transformation was performed on data that do not satisfy the assumptions of normality and homogeneity of variances. Repeated measures analysis of variance (ANOVA) was used to examine the effects of N addition and drought on Rs and soil moisture. Two-way ANOVA was used to test the effects of N addition, drought, and their interaction on soil properties, Rs, annual mean Rs, biomass, specific respiration, SOC, MBC, and DOC content. Individual treatment means within each moisture under the same N treatments and individual treatment means within each N treatment under the same moisture levels were compared with one-way ANOVA. We used the correlation analysis to explore the relationships between Rs, SOC, and biomass. All analyses were conducted using SAS software 9.2 (SAS Institute Inc., Cary, NC, USA) and statistical significance was determined at $p < 0.05$.

## 3. Results

### 3.1. Soil Properties

Soil temperature of different treatments showed strong seasonal patterns with higher temperature from June to August (Figure 1a). Both N addition and drought did not affect soil temperature. Soil moisture was significantly affected by the drought treatments ($p < 0.0001$, Figure 1b). The average soil moisture of well-watered, moderate drought,

and severe drought treatments were $0.282 \pm 0.013$ m$^3$ m$^{-3}$, $0.220 \pm 0.017$ m$^3$ m$^{-3}$, and $0.153 \pm 0.010$ m$^3$ m$^{-3}$, respectively.

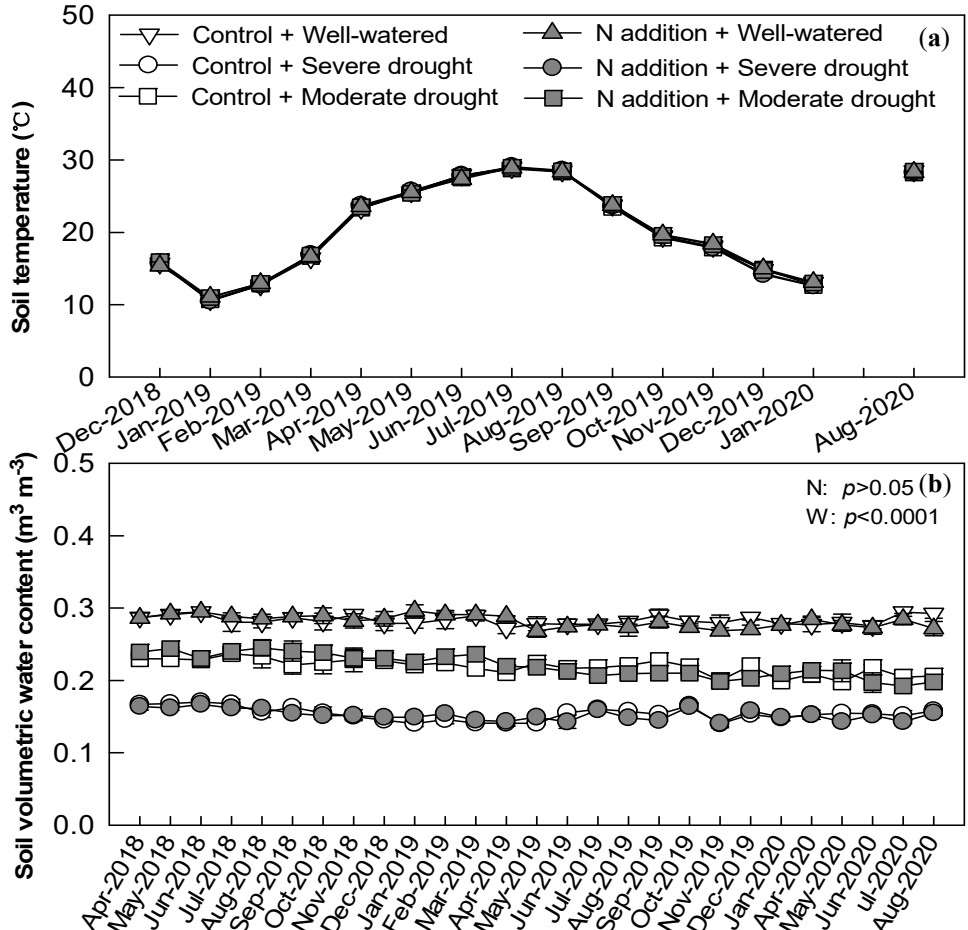

**Figure 1.** Soil temperature (**a**) and soil volumetric water content (**b**) in the different treatments.

N addition, drought, and their interaction affected ($p < 0.05$) soil pH, NO$^-_3$-N, and NH$^+_4$-N concentrations (Table 1). Overall, N addition decreased soil pH ($p < 0.001$) and AP concentrations ($p < 0.01$) and increased total N ($p < 0.05$), NO$^-_3$-N ($p < 0.001$), and NH$^+_4$-N concentrations ($p < 0.05$). Drought decreased soil pH ($p < 0.001$) and increased NO$^-_3$-N ($p < 0.01$) and NH$^+_4$-N concentrations ($p < 0.05$). In the well-watered and moderate drought conditions, N addition had little impact on soil pH, NH$^+_4$-N, and NO$^-_3$-N concentrations. However, in the severe drought condition, N addition significantly decreased soil pH and significantly increased NH$^+_4$-N and NO$^-_3$-N contents. N addition significantly increased total N in moderate drought treatment and significantly decreased AP in the well-watered treatment. Meanwhile, under the N addition groups, severe drought significantly decreased pH but significantly increased NO$^-_3$-N and NH$^+_4$-N concentrations, compared to well-watered and moderate drought treatments.

### 3.2. Soil Respiration

Rs was affected ($p < 0.05$) by N addition, drought, and their interaction (Figure 2). Rs rates was highest in July under different treatments. N addition increased ($p < 0.01$) Rs and annual mean Rs ($p < 0.05$). On the contrary, drought decreased ($p < 0.001$) both Rs and annual mean Rs. In the well-watered condition, N addition significantly increased annual mean Rs by 35.7%. However, in the moderate and severe drought condition, N addition did not affect Rs and annual mean Rs. In the control group, moderate drought did not affect annual mean Rs, but severe drought treatment significantly decreased annual

mean Rs by 61.5%. However, in N addition group, moderate drought and severe drought significantly decreased annual mean Rs by 27.6% and 70.5%, respectively.

**Table 1.** Effects of experimental nitrogen addition and drought on soil pH, total N, $NO^-_3$-N, $NH^+_4$-N, and available phosphate (AP) contents.

| Treatment | pH | Total N (g kg$^{-1}$) | NO$^-_3$-N (mg kg$^{-1}$) | NH$^+_4$-N (mg kg$^{-1}$) | AP (mg kg$^{-1}$) |
|---|---|---|---|---|---|
| Well-watered | | | | | |
| Control | 6.18 ± 0.14 [a] | 0.37 ± 0.02 | 1.99 ± 0.23 | 2.20 ± 0.57 | 5.42 ± 0.55 [A] |
| N addition | 6.02 ± 0.16 [a] | 0.39 ± 0.03 | 2.55 ± 0.84 [b] | 1.74 ± 0.50 [c] | 3.48 ± 0.80 [B] |
| Moderate drought | | | | | |
| Control | 5.98 ± 0.04 [b] | 0.37 ± 0.01 [B] | 2.56 ± 1.99 | 1.90 ± 0.42 | 4.60 ± 0.66 |
| N addition | 5.86 ± 1.18 [a] | 0.42 ± 0.03 [A] | 4.20 ± 0.40 [b] | 2.57 ± 0.43 [b] | 3.70 ± 0.06 |
| Severe drought | | | | | |
| Control | 5.98 ± 0.11 [bA] | 0.38 ± 0.04 | 3.92 ± 1.42 [B] | 2.28 ± 0.86 [B] | 5.79 ± 1.31 |
| N addition | 5.44 ± 0.03 [bB] | 0.41 ± 0.04 | 12.17 ± 5.27 [aA] | 3.96 ± 0.51 [aA] | 4.41 ± 0.31 |
| Analysis of variance (*p* values) | | | | | |
| N | <0.001 | 0.034 | <0.001 | 0.035 | 0.002 |
| W | <0.001 | 0.800 | 0.003 | 0.011 | 0.109 |
| N × W | 0.213 | 0.621 | 0.035 | 0.317 | 0.220 |

Different superscript indicates significant differences between N addition treatments under the same drought treatment (uppercase letters) and different drought treatments under the same N addition treatment (lowercase letters). Statistical significance was determined at *p* < 0.05.

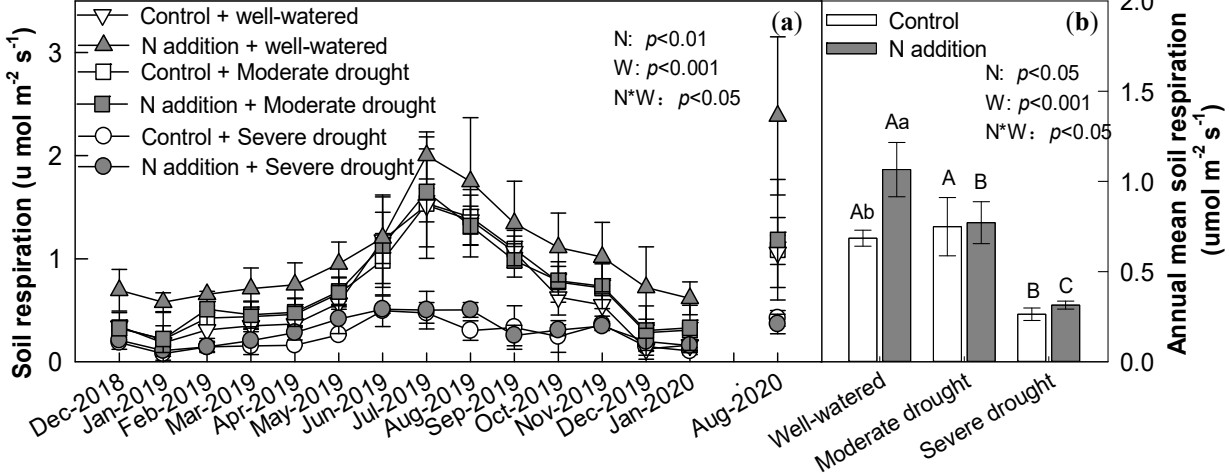

**Figure 2.** Soil respiration emission (**a**) and annual mean soil respiration emission (**b**) from January to December of 2019. The error bar represented arithmetic means ± standard errors for three replicates. Different superscript indicates significant differences between N addition treatments under the same drought treatment (lowercase letters) and different drought treatments under the same N addition treatment (uppercase letters). * represents interactions.

### 3.3. Aboveground Biomass and Belowground Biomass

Aboveground and belowground biomass were affected (*p* < 0.01) by N addition, drought, and their interaction (Figure 3). Overall, N addition increased (*p* < 0.001) both aboveground and belowground biomass. Drought decreased (*p* < 0.001) aboveground and belowground biomass, especially in the severe drought treatment. Under all three soil moisture conditions, N addition significantly increased aboveground biomass. In the well-watered and moderate drought conditions, N addition significantly increased belowground biomass. However, in the severe drought condition, N addition did not affect belowground biomass. Meanwhile, in the N addition group, severe drought decreased aboveground and belowground biomass, compared to moderate and severe drought.

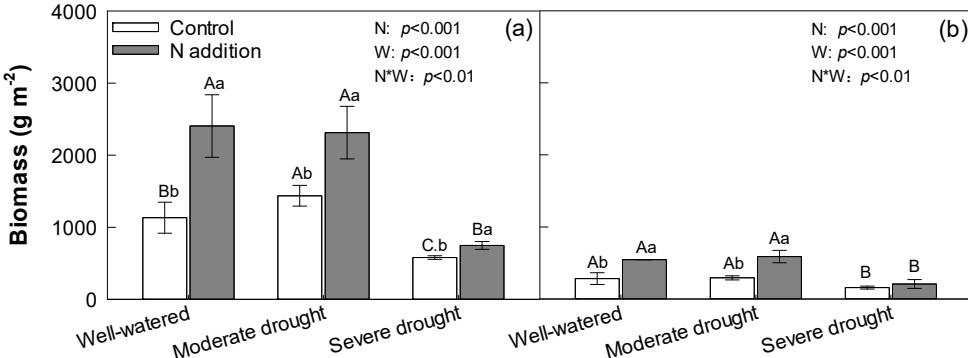

**Figure 3.** Aboveground (**a**) and belowground (**b**) biomass under different treatments. The error bar represented arithmetic means ± standard errors for three replicates. Different superscript indicates significant differences between N addition treatments under the same drought treatment (lowercase letters) and different drought treatments under the same N addition treatment (uppercase letters). * represents interactions.

### 3.4. Soil Organic Carbon, Microbial Carbon, and Dissolved Organic Carbon

N addition significantly increased ($p < 0.05$) SOC content in the 0–10 cm and marginally significantly increased ($p = 0.06$) SOC content in the 10–20 cm (Figure 4). In moderate drought condition, N addition significantly increased SOC content. Whereas in the N addition group, severe drought significantly decreased SOC content (Figure 4a). A similar tendency occurred in the 10–20 cm soil layer, where severe drought reduced the SOC content significantly, compared to the well-watered condition (Figure 4b).

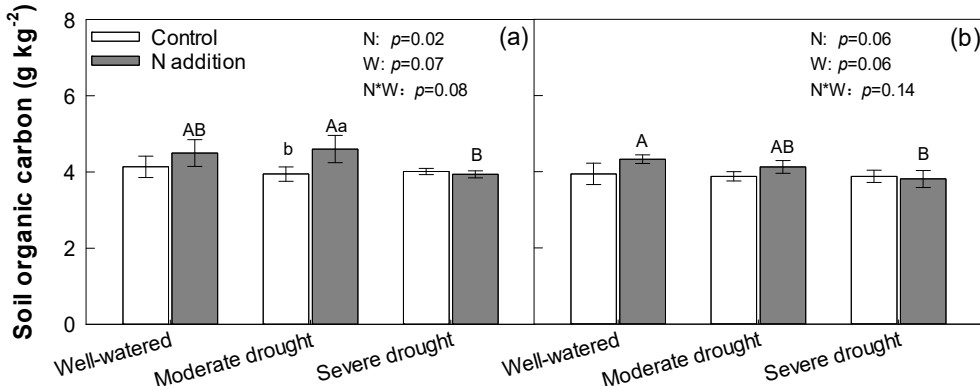

**Figure 4.** Soil organic carbon content in the 0–10 cm (**a**) and 10–20 (**b**) cm deep under different treatments. The error bar represented arithmetic means ± standard errors for three replicates. Different superscript indicates significant differences between N addition treatments under the same drought treatment (lowercase letters) and different drought treatments under the same N addition treatment (uppercase letters). * represents interactions.

Overall, N addition increased specific respiration ($p < 0.05$, Figure 5) and fine root biomass ($p < 0.01$). Drought decreased MBC content ($p < 0.05$), specific respiration ($p < 0.001$), and fine root biomass ($p < 0.001$). In the severe drought treatment, N addition significantly decreased MBC content. Meanwhile, in the control group, severe drought significantly decreased specific respiration and fine root biomass. In the N addition group, MBC content and fine root biomass were lower in the severe drought than in the moderate drought and well-watered conditions, soil DOC content was higher in the moderate drought than that in the severe drought and well-watered condition, and specific respiration significantly decreased with the reduction in soil moisture (Figure 5).

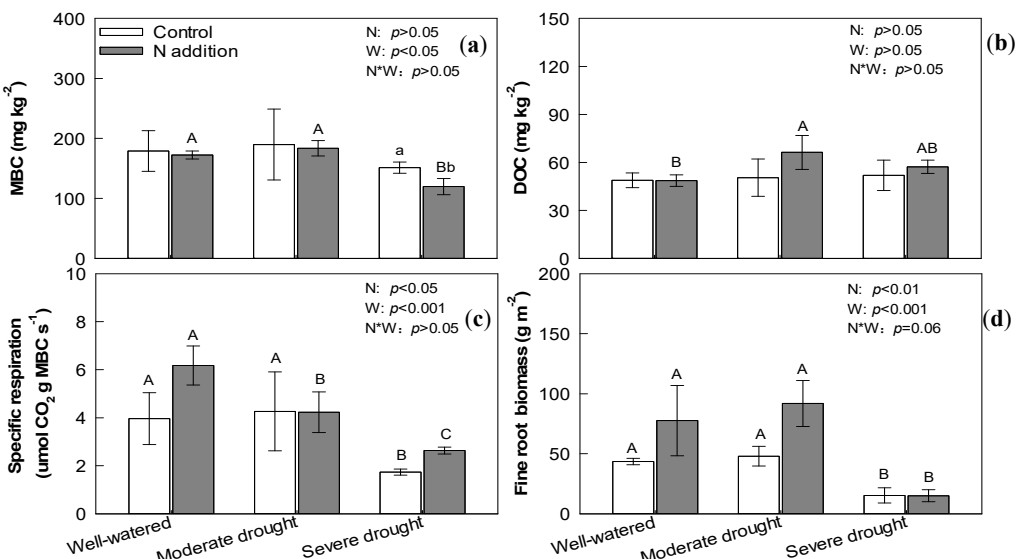

**Figure 5.** Soil microbial carbon (MBC) content (**a**) and dissolved organic carbon (DOC) content (**b**), specific respiration (**c**), and fine root biomass (**d**) under different treatments. The error bar represented arithmetic means ± standard errors for three replicates. Different superscript indicates significant differences between N addition treatments under the same drought treatment (lowercase letters) and different drought treatments under the same N addition treatment (uppercase letters). * represents interactions.

### 3.5. Relationships

Overall, we observed that Rs and SOC were highly significantly correlated with above- and belowground biomass and fine root biomass by correlation analysis (Figure 6). There were significant positive correlations ($p < 0.0001$) between Rs and either above-ground biomass, belowground biomass, or fine root biomass. Similar positive correlations ($p < 0.001$) were detected between SOC and aboveground biomass, belowground biomass, and fine root biomass.

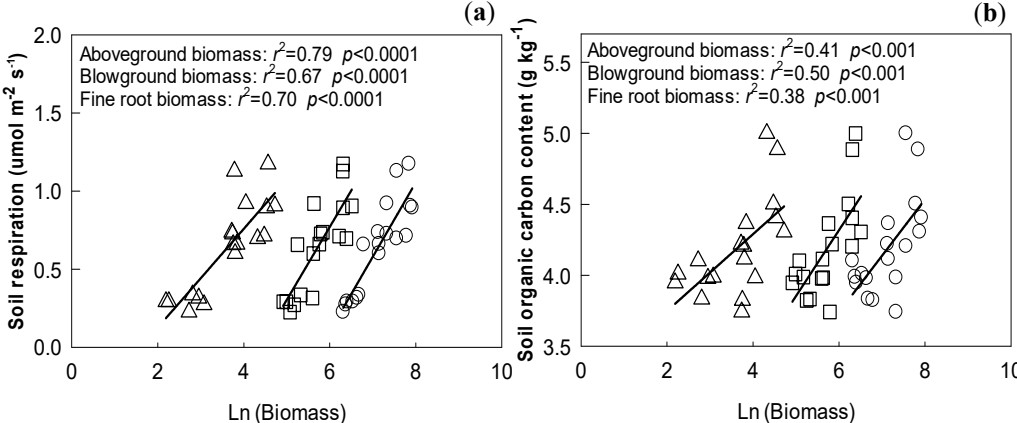

**Figure 6.** Relationship between soil respiration (**a**) or soil organic carbon content (**b**) and biomass. Triangle, square, and circle represented aboveground biomass, belowground biomass and fine root biomass, respectively.

## 4. Discussion

### 4.1. Effect of N Addition, Drought, and Their Interaction on Soil Respiration

Many studies reported that N deposition inhibited Rs [19,20]. However, we found that N addition promoted Rs in this study (Figure 2). There are also some studies that found that N addition enhance Rs; this may be related to plant growth and soil microbes [22,43]. We observed that N addition increased fine root biomass (Figure 5d), which may promote

Rs by increasing root respiration [14]. In addition, N availability may affect soil microbial biomass and activity, which further affect Rs. For example, Zhang et al. [44] showed that an increase in soil N availability promoted soil microbial biomass and activity by reducing the impact of N limitation on microorganisms. In our study, although N addition did not significantly affect MBC content (Figure 5a), it increased specific respiration (Figure 5c), which indicated N addition may stimulate microbial activity, and thus increase Rs.

Most previous observations indicated that drought decreased Rs [9,45]. This is partially in agreement with our study; in our study, severe drought significantly inhibited Rs (Figure 2). Poorter et al. [46] reported that drought can alter the allocation of assimilates to the root system and to the respiration by roots and their symbionts. We found that severe drought-inhibited fine root biomass (Figure 5d), the possible reason may be that plants altered photosynthetic C allocation, promoted the production of root secretions, and reduced the amount of C allocated by biomass production and Rs under drought condition [47,48]. On the other hand, soil moisture could influence Rs through physiological processes of soil microorganisms [11,16]. Soil microbial activity and soil moisture are closely related. In our study, severe drought decreased MBC content and specific respiration (Figure 5a,c). Thus, the reduction in soil moisture might inhibit the quantities and activity of soil microbes under severe drought and might decrease the soil microbial respiration.

In our study, N addition promoted Rs, but drought offset this promotion effect (Figure 2), which differed to hypothesis one that N addition would inhibit Rs. We found N addition and drought have antagonistic non-additive effects on Rs. In general, the effect of the interaction of N addition and drought on Rs varies with soil moisture [25,27]. First, the reduction in soil moisture might decrease the mobility of the soil available N [24]. As soil moisture decreased, soil available N mobility was inhibited, which may weaken the effects of N addition on plant growth, and thus decrease the roots growth and Rs [28,49]. Our result that the positive correlation between Rs and aboveground biomass, belowground biomass, and fine root biomass (Figure 6a) could also support this result. This is because the increased aboveground biomass could provide continuous C source for soil microbial decomposition, as well as the increased belowground biomass and fine root biomass could promote root respiration. In our study, drought limited the positive effect of exogenous N on plant growth (aboveground biomass, belowground biomass, fine root biomass), which may lead to inhibition of Rs. In addition, the effect of N addition on microbial activity might strongly depend on the variation of soil moisture [31]. We found that N addition promoted specific respiration (Figure 5c), but severe drought decreased specific respiration in the N addition group ($p < 0.0001$). Thus, these results indicated that drought could offset the promotion effect of N addition on specific respiration. Previous studies reported that sufficient water could enhance the effect of N deposition on soil microbial community and activity [50], which suggested that drought may inhibit the impact of N addition on soil microbial activity, further decreasing Rs [26]. Thus, in order to explore the effect of N deposition on forest Rs, it is necessary to consider soil moisture effects under the background of climate change.

We also observed that under the N addition group, the inhibition effect of Rs by drought was more visible, compared with the control group (Figure 2). The possible reason was that N addition increased fine root biomass (Figure 5d), further increasing the demand for soil moisture in order to sustain the plant activities. Meanwhile, other studies showed that N addition affected plant demand for soil moisture [51], and changes in soil moisture might further affect Rs [9,30,45]. Therefore, under the context of aggravated nitrogen deposition, the inhibitory effect of drought on soil respiration will be more obvious in the future.

### 4.2. Effect of N Addition, Drought, and Their Interaction on Soil Organic Carbon

Previous studies show that N addition might inhibit the decomposition of soil microbes, further enhancing the soil C sequestration [12,21,52]. In this study, we also found that N addition increased SOC content (Figure 4). Due to SOC content intensively de-

pending on the balance between C input and its decomposition, our result indicated that the increasing of SOC content might be caused by the amount of plant C input that was higher than the C output by soil respiration. Most of the studies indicated that the roots system was an essential source of SOC content [53,54]. Meanwhile, Wang et al. [54] showed that root biomass (or belowground biomass) was positively correlated with SOC content. In our study, the variation of belowground biomass and fine root biomass was in line with SOC content (Figure 6b), which to some extent showed that the increasing of SOC content induced by N addition was associated with the increasing of roots biomass. In addition, many studies revealed that N addition promoted SOC content by inhibiting the process of microbial decomposition [12,52]. However, other research also indicated that N input may stimulate soil microbes, further decreasing soil C sequestration by alleviating N limitation [19,43,44]. We found that in our N-limited environment (Table 1), N addition increased specific respiration and Rs (Figures 2 and 5c) and also increased SOC content (Figure 4). These results indicated that the amount of plant C input is higher than C output, and thus leading to the increased SOC content in the N addition group.

In our study, drought slightly decreased SOC content (Figure 4), which is similar to previous studies, which found drought decreased SOC, and thus affected C sequestration [10,55]. We considered that the root was an important cause of SOC content altering in this study (Figure 6b). Severe drought significantly decreased belowground biomass and fine root biomass (Figures 3b and 5d), which might lead to the reduced contribution of root to SOC accumulation. Our result was agreement with some studies that drought would inhibit soil C stock by affecting belowground biomass [10,56]. Meanwhile, the reduction in aboveground C input driven by drought may also be partly responsible for the impact on soil C stock. Some studies also showed that drought affected aboveground C process, and decreased the aboveground biomass input, further decreasing SOC accumulation [55,57]. The reduction in aboveground biomass in the severe drought treatment was also observed in our study (Figure 3a).

We found that N addition increased SOC content, but severe drought would offset this effect (Figure 4), which was in agreement with the second hypothesis. Likewise, Xiang et al. [58] found that water reduction reduced SOC content under the same stimulated N deposition conditions. Roots was an important source of SOC [53,54]. We also found that N addition increased belowground and fine root biomass, but severe drought would inhibit this effect (Figures 3b and 5d), which could lead to severe drought inhibiting the positive effect of N addition on SOC content. This above result could be supported by the positive correlation between SOC and aboveground and belowground fine root biomass (Figure 6b). In this study, N addition increased the belowground biomass and fine root biomass that may increase root secretion and the ability of soil C sequestration and could also directly promote plant-derived C input (e.g., litterfall, etc.), which may further increase SOC content. However, drought inhibited the positive effect of N on above- and belowground biomass and fine root biomass, which could further offset the promotive effect of N addition on SOC content. In addition, drought might indirectly strengthen the inhibitory effect of N addition on MBC content. We observed that N addition inhibited MBC content in the severe drought condition (Figure 5a), which may be partly the reason for the suppression of SOC content. Moreover, the other possible reason was that soil aggregation instability may affect SOC sequestration. Chen et al. [59] showed that the interactive effect of N addition and moisture decreases soil aggregate stability through reducing biological binding agents. Thus, we suspected that the interaction of N addition and severe drought reduced MBC, DOC, fine root biomass (Figure 5), and other binding agents, leading to the destabilization of soil aggregates and resulting in loss of SOC content.

Meanwhile, we also observed that under the N addition group, the inhibition effect of SOC content by drought was more obvious when compared to the control group (Figure 4). Högberg et al. [60] showed that N addition might disproportionately increase leaf biomass, further increasing the probability of insufficient water supply, which may enhance the susceptibility of plant to N addition in the background of drought [51]. Under the condition

of the plant's demand for water, which was caused by N addition, soil moisture changes might further affect soil C stocks [10,55]. Therefore, the suppressive effect of drought on SOC content would be more obvious in the context of aggravated N deposition in the future.

## 5. Conclusions

In this study, we found that N addition promoted Rs, whereas drought inhibited the promoting effect of Rs by N addition. Consistent with changes with Rs, N addition promoted aboveground biomass, belowground biomass, fine root biomass, and specific respiration, but severe drought offset these promotive effects. Both severe drought and N addition inhibited MBC content. Moreover, we found that N addition enhanced SOC content, but severe drought offset the promotion of SOC content by N addition. Notably, both Rs and SOC content were significantly and positively correlated with fine root biomass, aboveground biomass, and belowground biomass. We also found similar changes in Rs and SOC content, possibly due to the dominance of C input to SOC and Rs contribution in young N limited forests. In the context of future N deposition, the inhibited effect of drought on Rs and C capture may be more significant in subtropical forest. Our findings highlight the need to focus on the effects of multi-factor interactions on forest Rs and C sequestration under the background of global change.

**Author Contributions:** X.F. conceived and designed the experiment; Y.-l.Z. and X.-p.L. completed the experiment; Z.-g.Y., Y.-p.H. and X.-h.T. analyzed the experimental data; Y.-l.Z. wrote the paper. All authors have read and agreed to the published version of the manuscript.

**Funding:** This research was funded by the National Science Foundation of China (Grant Nos. 41907274, 42267034, and 41877326) and the Key Science and Technology Research Project of Fujian Pine Forest Reconstruction and Upgrading Action (2021FKJ01).

**Institutional Review Board Statement:** Not applicable.

**Informed Consent Statement:** Not applicable.

**Data Availability Statement:** Not applicable.

**Acknowledgments:** The authors also thank Yalin Hu for his help with field work.

**Conflicts of Interest:** The authors declare no conflict of interest.

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
