# Peer review of "Drought Offsets the Potential Effects of Nitrogen Addition on Soil Respiration and Organic Carbon in Model Subtropical Forests"

_forests, doi:10.3390/f13101615_

Round 1
Reviewer 1 Report
The manuscript is of interest to the readers of the journal and the topic is a hot area with multiple environmental change factors. I would suggest the manuscript needs minor revision to improve the language. Scientifically, the authors have done a good job.
-No leachate measurement?
-Some grammar issue below
"However, there is no clear how forest Rs is affected by combin-ing N deposition with drought"
"However, there is limited about understanding the potential effects of N and drought interaction on Rs and C sequestration in subtropical forests."
"Therefore, we conducted a 2.5-year regulation experiment," "regulation" to "manipulation"
" N addition may stimulus soil organic carbon accumulation by reducing the process of soil C decom-position" stimulate
"One drainage port (2 cm) at the bottom of each square pond con-nected to a polyvinyl chloride tube to collect the soil leachate." was connected
"In April 2018, the experiment was started with two groups of N addition treatments" was initiated, two levels of N addition
"Fine root biomass was calculated by fine roots which was collected during soil sieving." Fine roots collected during soil sieving was used for fine root biomass.
"Five soil cores (4.5 cm in diameter) of within each plot were randomly collected from at 0-10 cm and 10-20 cm deepth"
"All samples were pass through a 2 mm sieve to picked out fine roots roots, litter and plants residues, and the fine roots were used to calculate fine root biomass."
"and the second one was stored at 4 °C for the later soil microbial biomass determination"
"Soil ammonium nitrogen (NH+4-N) and nitrate nitrogen (NO-3-N) was determined by"
"conversion coefficients were was 0.45 for MBC"
"Overall, we observed that soil respiration and SOC were highly significantly correlated with above- and belowground biomass and fine root biomass by correlation analysis"
Author Response
- Responses to the Reviewer #1’s comments
- No leachate measurement?
Response : Yes. We did not measure and analyse the soil leachate. Thank you for the comment, we will pay attention to the leachate in our subsequent study.
- Grammar issue : However, there is no clearhow forest Rs is affected by combining N deposition with drought.
Response : Thanks. We have reworded this sentence. Please see line 47.
- Grammar issue :However, there is limited about understanding the potential effects of N and drought interaction on Rs and C sequestration in subtropical forests.
Response: Thanks. We have revised this sentence. Please see line 94.
- Grammar issue: Therefore, we conducted a 2.5-year regulation experiment," "regulation" to "manipulation"
Response: Done. We have made the change accordingly. Please see line 96.
- Grammar issue: N addition may stimulussoil organic carbon accumulation by reducing the process of soil C decomposition. stimulate
Response: Done. Thanks, please see line 101.
- Grammar issue:One drainage port (2 cm) at the bottom of each square pond connected to a polyvinyl chloride tube to collect the soil leachate. was connected
Response: Done. We have made the change accordingly. Please see line 112-113.
- Grammar issue:In April 2018, the experiment was started with two groups of N addition treatments. was initiated, two levels of N addition
Response: Thank you. We have revised this sentence. Please see line 122.
- Grammar issue :Fine root biomass was calculated by fine roots which was collected during soil sieving." Fine roots collected during soil sieving was used for fine root biomass.
Response: Thanks. We have re-written this sentence. Please see line 159-161.
- Grammar issue: Five soil cores (4.5 cm in diameter) ofwithin each plot were randomly collected from at 0-10 cm and 10-20 cm deepth.
Response: Done. We have revised this sentence. Please see line 164.
- Grammar issue: All samples were passthrough a 2 mm sieve to picked out fine roots roots, litter and plants residues, and the fine roots were used to calculate fine root biomass
Response : Thank you. We have made the change accordingly. Please see line 166-167.
- Grammar issue: and the second one was stored at 4 °C forthe later soil microbial biomass determination
Response: Done. Thanks. Please see line 169.
- Grammar issue: Soil ammonium nitrogen (NH+4-N) and nitrate nitrogen (NO-3-N) wasdetermined by.
Response: Done. Thanks. Please see line 172.
- Grammar issue : conversion coefficientswere was45 for MBC
Response: Done. Thank you. Please see line 178.
- Grammar issue :Overall, we observed that soil respiration and SOC were highly significantly correlated with above- and belowground biomass and fine root biomass by correlation analysis.
Response: Done. Thanks. Please see line 289-290.
Reviewer 2 Report
1. What is the basis for setting the parameters of nitrogen addition and water drought, such as 80%, 60%, 40%?
2. Is this study an indoor simulation experiment? How is in situ soil collected? How to ensure that the soil properties are approximate?
3. Figure 1,Soil temperature can not be seen clearly, it is recommended to enlarge the scale or change to a table; Soil water content is also recommended to be scaled up.
4. Table 1, three N addition treatments, how to distinguish?
5. Figure 2, it is also recommended that the appropriate optimization, scale up some can be solved.
6. In 4.1 (discussion),Please discuss why biomass and respiration rate are these relationships (Figure 6)?
7. It is suggested to discuss more about the offset effect of water deficit on N addition.
Author Response
- Responses to the Reviewer #2’s comments
- What is the basis for setting the parameters of nitrogen addition and water drought, such as 80%,60%, 40%?
Response: Thanks. The amount of N deposition in South China can reach up to 40 kg N ha-1 y-1, so we set the N addition to 80 kg N ha-1 y-1, which is the double of the N deposition in South China. In addition, ca. 40% of field capacity represents the extreme dry condition, ca. 60% of field capacity represents the slightly dry condition, and ca. 80% of field capacity represents the water sufficiency conditions. However, it is our experimental design, the detail soil moisture detection was shown on Figure 1. The average soil moisture of well-watered, moderate drought, severe drought treatments was were 0.282±0.013 m3 m-3, 0.220±0.017m3 m-3, 0.153±0.010 m3 m-3, respectively.
- Is this study an indoor simulation experiment? How is in situ soil collected? How to ensure that the soil properties are approximate?
Response: Yes. This study is an indoor manipulation experiment. However, we did not collect situ soil. Two different soil layers (0-25, 25-50) were collected in a nearby secondary natural forest in November 2017, mixed separately after picking out coarse roots and stones, and then filled into the corresponding plot (0.5 m depth). After approximately 2 months of soil natural sinking, four local typical plants species were transplanted into each plot randomly. We also studied the effect of environmental factor changes on soil organic carbon by the same method in the previous study, for example Fang et al. 2020.
(Fang, X.; Zhou, G.Y.; Qu, C.; Huang, W.J.; Zhang, D.Q.; Li, Y.L.; Yi Z.G.; Liu, J.X. Translocating subtropical forest soils to a warmer region alters microbial communities and increases the decomposition of mineral-associated organic carbon. Soil Biol. Biochem., 2020, 142:107707)
- Figure 1,Soil temperature can not be seen clearly, it is recommended to enlarge the scale or change to a table; Soil water content is also recommended to be scaled up.
Response: Done. Thank you. We have scaled up the Figure 1. Please see line 203-204.
- Table 1, three N addition treatments, how to distinguish?
Response: Thank you for the comment. We set up at two levels of N addition treatments (control and N addition) and three levels of moisture (well-watered: ca. 80% of field capacity, moderate drought: ca. 60% of field capacity, severe drought: ca. 40% of field capacity). This means that there are three moisture gradient interaction treatments under each N addition group, with a total of six cultivation plots.
- Figure 2, it is also recommended that the appropriate optimization, scale up some can be solved.
Response: Thanks. We have scaled up the Figure 2. please see line 237.
- In 4.1 (discussion),Please discuss why biomass and respiration rate are these relationships (Figure 6)?
Response: Thank you for the comment. Soil respiration contains both autotrophic respiration and heterotrophic respiration. Therefore, an increased aboveground biomass can provide C for soil microbial decomposition, which further promotes soil respiration. And increased belowground biomass and fine root biomass could contribute to increased root respiration, which further has a positive effect on soil respiration. We made some additions, please see the line 333-338.
- It is suggested to discuss more about the offset effect of water deficit on N addition.
Response: Thank you for the suggestion. N addition increased plant growth and decomposition sources for microbes, which may increase soil respiration and SOC content. However, drought inhibited the availability and mobility of soil N, which could limit the growth of plant and thus decrease the plant-derived C input, further decrease soil respiration and SOC content. In addition, severe drought inhibited soil microbial activity, weaken the deposition of soil carbon and the use of exogenous N by soil microbiology, which may lead to the inhibition of soil respiration and SOC content. We made some additions, please see line 333-338 and line 396-401.